# Observation Model for Indoor Positioning

**DOI:** 10.3390/s20144027

**Published:** 2020-07-20

**Authors:** Berthold K. P. Horn

**Affiliations:** Department of Electrical Engineering and Computer Science, MIT, Cambridge, MA 02139, USA; bkph@csail.mit.edu

**Keywords:** Bayesian grid, observation model, transition model, indoor position, indoor location, relative permittivity, fine time measurement, round trip time, FTM, RTT, IEEE 802.11mc, IEEE 802.11–2016

## Abstract

The IEEE 802.11mc WiFi standard provides a protocol for a cellphone to measure its distance from WiFi access points (APs). The position of the cellphone can then be estimated from the reported distances using known positions of the APs. There are several “multilateration” methods that work in relatively open environments. The problem is harder in a typical residence where signals pass through walls and floors. There, Bayesian cell update has shown particular promise. The Bayesian grid update method requires an “observation model” which gives the conditional probability of observing a reported distance given a known actual distance. The parameters of an observation model may be fitted using scattergrams of reported distances versus actual distance. We show here that the problem of fitting an observation model can be reduced from two dimensions to one. We further show that, perhaps surprisingly, a “double exponential” observation model fits real data well. Generating the test data involves knowing not only the positions of the APs but also that of the cellphone. Manual determination of positions can limit the scale of test data collection. We show here that “boot strapping,” using results of a Bayesian grid update method as a proxy for the actual position, can provide an accurate observation model, and a good observation model can nearly double the accuracy of indoor positioning. Finally, indoors, reported distance measurements are biased to be mostly longer than the actual distances. An attempt is made here to detect this bias and compensate for it.

## 1. Background

There has been considerable interest in developing the ability to accurately localize position indoors where GPS can not be used [1,2,3,4,5,6,7,8,9,10,11]. The IEEE 802.11mc WiFi standard provides a protocol for an initiator (cellphone) to estimate its distance from responders (WiFi APs) [12,13,14,15,16,17,18,19,20]. Actually, what is reported is half of the round-trip time (RTT) of an RF signal, multiplied by the speed of light. This *would* be the distance if the signal travelled in vacuum or air in a straight line (and ignoring noise).

Unfortunately, indoors, this simple picture does not reflect reality. First, the estimate of the time of flight is corrupted significantly by a large *position-dependent* error [21] (which may be due to interaction of super-resolution algorithms with fast fading patterns of signal strength) [22].

Secondly, indoors, WiFi signals are slowed down by travel through building materials of high relative permittivity [23,24,25,26,27] (The distance added by an obstacle of thickness *d* and relative permittivity ε is $d(\sqrt{\varepsilon }-1)$). Finally, in some cases, the direct line of sight is blocked by metallic objects or thick layers of absorbing material. In this case the round-trip-time estimate may be based on a signal that has reflected off some surface away from the direct path. In either case, the reported “distance” is larger than the actual distance. Since there is significant bias in the reported distances, one should not expect good results in “multilateration” if one treats reported “distances” as if they were the actual distances.

Of various methods for estimating the position of the initiator, the Bayesian grid update method [28] has shown promise [21]. Bayesian grid update requires (i) a transition model, and (ii) an observation model. The transition model describes how the initiator may move in the time interval between measurements. The observation model gives the probability of observing a particular *reported* distance conditioned on the *actual* distance between initiator and responder. We show here that a parameterized “double exponential” model fits real data well. Results of using this observation model in Bayesian cell updates are shown in Figure 1 (screen shots from the video [29]).

The accuracy of distance measurement using FTM (Fine Time Measurement) RTT (Round Trip Time) (as per IEEE 802.11mc) is adequate for some applications but not others [21]. Frequency diversity makes it possible to double the accuracy [21], but this is not supported by the current cellphone application programming interface (API). Consequently, increasing the accuracy of position determination by constructing good observation models is important.

We explore the question of the observation model first in a “2-D” setting (single level of a large house), and then in a “3-D” setting (three levels in another house).

## 2. Observation Model

The observation model gives the probability of observing a particular distance conditioned on the actual distance between initiator and responder. If there were no errors, the observation model would be an impulse where reported distance equals actual distance. In the presence of Gaussian measurement noise, the impulse would be spread out into a Gaussian of appropriate standard deviation.

Due to the biases described above, however, an observation model is quite asymmetrical with respect to the peak value. Ignoring effects due to measurement errors for the moment, the reported distance would always be equal to or greater than the actual distance. In the presence of measurement errors, we can expect *some* reported distances to be smaller, but do still expect a rapid drop off from the peak where the reported distance is smaller than the actual distance. Conversely, due to the bias, we can expect a *slow* drop off where the reported distance is larger than the actual distance.

## 3. Building an Observation Model Using Experimental Measurements (2-D Case)

An observation model can be based on scattergrams of reported (observed) distance for known actual distances. We can obtain NM data points if we have *N* responders in known positions and take measurements of distances from *M* known positions for the initiator (see e.g., Figures 8 and 9 in [21]). Here we start with a planar “2-D” example of a single level of a large house. The scattergram in Figure 2 shows reported distance versus actual distance. (The average measurement offset of this initiator/responder system has been determined and subtracted out [21]). Such scattergrams can be used to estimate parameters of the underlying probability distribution.

Conveniently, experimental results indicate that the scattergram has structure that allows reduction of what appears to be a two-dimensional probability density distribution fitting problem to a one-dimensional one. In particular, note that in the above scattergram, the vertical spread in reported distances increases more or less in proportion to the actual distance.

This regularity becomes more obvious if we plot the *ratio* of reported distance to actual distance (versus actual distance), as in Figure 3. The vertical scatter in this diagram is more or less independent of actual distance (except perhaps for very small and very large distances). So we can collapse the two-dimensional scattergram into one were we plot the *ratio* of distances versus the actual distance. It is, of course, easier to fit one-dimensional distributions than two-dimensional distributions, particularly when the number of available samples is limited. This approach has been used before to obtain a piece-wise linear approximation of an observation model from relatively few measurements (see Figure 10 in [21]).

The shape of the histogram becomes clearer when we have many measurements. Figure 4 shows a histogram of the ratio of reported to actual distance as a function of actual distance for about 10,000 measurements (collected in about 10 min). These measurements are from a single level of a house of mostly wooden stud and dry wall construction, but with some heavily tiled bathroom walls, some concrete support beams, and metallic obstacles such as a large refrigerator, an oven and stacked microwave. Here, the asymmetry is quite apparent, with ratios greater than one occurring much more frequently than ratios smaller than one (From the figure that follows we see that about 85% of the ratios are larger than one in this case).

Replotting the histogram on a logarithmic scale, as in Figure 5, shows that the drop offs on either side of the peak are exponential. As expected, the decay is much more gradual for ratios larger than one than it is for ratios smaller than one.

A model fit to the histograms from these experimental results is the “double exponential”
(1)$$h\left(r\right)=\frac{1}{H}\left\{\begin{array}{cc} e^{-\left(1-r\right)/s_{l}} & \mathrm{for}\;r<1\\ e^{-\left(r-1\right)/s_{r}} & \mathrm{for}\;r>1 \end{array}\right.$$
where r=o/d is the ratio of, *o*, the reported (observed) distance, to, *d*, the actual distance (with $s_{r}>s_{l}$). The constant $H=s_{l}+s_{r}$ normalizes the distribution so its integral (w.r.t. *r*) equals one. For the solid curves plotted in Figure 4, Figure 5 and Figure 6, $s_{l}=0.033$ and $s_{r}=0.145$.

In terms of h(…), the conditional probability of reported distance, *o*, when the actual distance is *d*, is
(2)$$P\left(o|d\right)=\frac{1}{d}h\left(\frac{o}{d}\right)$$

Another check on the quality of the parametric fit is given by the cumulative distribution shown in Figure 6, where the dots are from experimental data while the solid curve is from the double exponential fit.

The parameters, $s_{l}$, $s_{r}$, estimated from the scattergram, depend on the structure of the building and the building materials. Fortunately, experiment indicate that they need not be known with great accuracy to get reasonable results with the Bayesian grid update method. The key is to get about the right amount of asymmetry.

An aside: In practice a weighted sum of this distribution and a uniform distribution should be used to account for the appearance of random outliers in the measurements.

The model could be further refined by breaking down the scattergram into sections based on distance, since the best-fit parameters seem to depend somewhat on distance. The best-fit $s_{l}$ appears to go down with distance, while $s_{r}$ may go up a bit. This refinement does not change the behavior of the Bayes cell update significantly, and is not needed to get good results.

## 4. Using Calculated Position of Initiator as a Proxy

We need to know the actual distances between initiator and responders when fitting a parameterized model to measurement data. Now, in order to use the method described here for indoor positioning, the positions of the responders already have to be known. We can calculate the actual distances between the initiator and each of the responders, as required for the model fitting, if the positions of the initiator while taking data are also known. While one can certainly set up an experimental situation where the position of the initiator is known accurately, this approach tends not to scale well (unless some independent method for automatically determining indoor position can be found). A manual measurement process as used in [21] is not very practical if we want tens of thousands of measurements.

It may make sense then to use a proxy for the true initiator position if a good enough one can be found. For this purpose, Bayesian grid update can be used to estimate the position of the initiator. The quality of this proxy measurement increases with the number of responders, so this is best done with more than the minimum number of responders needed to solve the multilateration problem In the 2-D case, for example, three responders are the minimum needed for unambiguous solution [21], but it is better to use more. Six responders are shown in Figure 1, which was the setup for data collection, when three responders would have been enough to cover most of the area for 2-D indoor positioning purposes. (Similarly, nine responders are shown, when four responders would have been enough for 3-D indoor positioning).

The resulting scattergram can be used to fine-tune the observation model. The new observation model can then be used in turn for another round to further refine the parameters of the observation model (no need to take new data).

Figure 7 shows slices through an observation model. Each curve shows the conditional probability of a particular measured distance given the actual distance. The plots are for actual distances from 1 to 20 m in 1 m increments.

### Accounting for Random Measurement Error

The estimate of the conditional probability (from Equation (Equation 2)) has peaks that get sharper the smaller the actual distance. In practice, additive random measurement error will smear out such peaks. This can be modelled by convolving the distribution with some local averaging kernel, such as a Gaussian (see Appendix D). The standard deviation of the kernel can be chosen equal to the reported standard deviation from repeated measurements taken in the same position. This “smearing” has little effect on conditional probabilities for longer distances, but will smooth out unrealistically sharp peaks that would otherwise occur for small distances. See e.g., Figure 8.

## 5. Efficient Use of the Observation Model in Bayesian Grid Update

The Bayesian grid update requires stepping through all of the cells of the grid and determining the conditional probability that one would observe the distance reported given the known actual distance of each cell from the responder.

In detail, the whole grid is updated when a reported distance measurement *o* is received relating to responder *i* at $x_{i}$. For each cell in the grid, corresponding to its position x, one first computes the distance from the responder $d=∥x-x_{i}∥$. The probability currently stored in the cell corresponding to x is then multiplied by the conditional probability P(o|d) (After sweeping through the grid in this fashion, the values can be renormalized so that they add up to one).

An aside: One must be careful not to let the limitations of floating-point multiplication produce an actual zero, since once a value is zero it cannot become non-zero again in the multiplicative Bayesian update process.

After the update, if a specific position is needed for the initiator, rather than a probability distribution, then one can use either the peak (maximum likelihood) or the centroid (expected value) of the distribution.

To speed up the above sweep, a “rate vector” can be precomputed when an observation becomes available and used for the whole grid. The rate vector gives the probability that the reported distance, *o* could correspond to the actual distance, *d*, that is,
(3)$$r_{o}\left(d\right)=P\left(o|d\right).$$

Precomputing this rate vector can speed up the Bayesian cell update if the computation of the conditional probability P(o|d) is non-trivial. This is particularly helpful in the case of a 3-D grid of voxels where the number of elements in the rate vector can be much smaller than the number of cells in the grid.

Figure 9 shows a set of “rate vectors” plotted versus actual distance for reported distances in the range 1 to 20 m in 1 m increments. (Note that the horizontal axis in Figure 7 shows reported distances, while the horizontal axis in Figure 9 shows actual distances).

As mentioned above, one should take into account the “smearing” effect of measurement errors on the conditional probabilities. Figure 10 shows the rate vector based on the smoothed conditional probability shown in Figure 8 (Note that this is *not* the same as convolving the rate vector itself with the smoothing kernel).

## 6. Building an Observation Model Using Experimental Measurements
(3-D Case)

Next, we consider a more complex situation where position is recovered in three dimensions in a building with three floors. We show here that a parameterized “double exponential” model with a flat top fits real data well. Results of using this observation model in Bayesian cell updates are shown in Figure 11 (screen shots from the video [30]).

The scattergram in Figure 12 shows reported distance versus actual distance. Again, experiments suggest that the scattergram has structure that allows reduction of what appears to be a two-dimensional probability density distribution fitting problem to a one-dimensional one.

This regularity becomes more obvious if we plot the *ratio* of reported distance to actual distance (versus actual distance). as in Figure 13. The vertical scatter in this diagram is more or less independent of actual distance (except perhaps for very small and very large distances). So we can collapse the two-dimensional scattergram into one were we plot the *ratio* of distances versus the actual distance.

Figure 14 shows a histogram of the ratio of reported to actual distance as a function of actual distance for about 20,000 measurements. These measurements are from three levels of a house of wooden stud and dry wall construction, with a brick chimney and metallic obstacles such as a large refrigerator, water heater, furnace, and a wood stove. The asymmetry is again apparent, with ratios greater than one occurring much more frequently than ratios smaller than one (we see that about 97% of the ratios are larger than one).

Replotting the histogram on a logarithmic scale, as in Figure 15, shows that the drop offs on either side of a flat top are exponential. As expected, the exponential decay is much more gradual on the left than on the right.

A model fit to the histograms from these experimental results is the “double exponential” with flat top
(4)$$h\left(r\right)=\frac{1}{H}\left\{\begin{array}{cc} e^{-\left(r_{l}-r\right)/s_{l}} & \mathrm{for}\;r<r_{l}\\ 1 & \mathrm{for}\;r_{l}\leq r\leq r_{r}\\ e^{-\left(r_{r}-1\right)/s_{r}} & \mathrm{for}\;r_{r}<r \end{array}\right.$$
where r=o/d is the ratio of, *o*, the reported (observed) distance, to, *d*, the actual distance. The distribution is constant in the region between the two decaying exponentials (i.e., for $r_{l}\leq r\leq r_{r}$). The “flat top” provides for a better fit in this more complex situation. It arises here because in this house the line of sight is rarely unobstructed and distance measurements go through varying amounts of material with high relative permittivity. For a start, in any given position, about 2/3 of the measurements are w.r.t. responders on other floors and thus go through one or two layers of flooring material.

The constant $H=s_{l}+\left(r_{r}-r_{l}\right)+s_{r}$ normalizes the distribution so its integral (w.r.t. *r*) equals one. For the solid blue curves plotted in Figure 14, Figure 15 and Figure 16, $s_{l}=0.045$, and $s_{r}=0.136$, while $r_{l}=1.07$ and $r_{r}=1.17$.

Again, the conditional probability of reported distance, *o*, when the actual distance is *d*, is
(5)$$P\left(o|d\right)=\frac{1}{d}h\left(\frac{o}{d}\right)$$

Another check on the quality of the parametric fit is given by the cumulative distribution shown in Figure 16, where the dots are from experimental data while the solid blue curve is from the double exponential fit with flat top. About 97% of the ratios of measured to actual distance are larger than one.

The best-fit parameters, $s_{l}$, $s_{r}$, $r_{l}$, and $r_{r}$, estimated from the scattergram, depend somewhat on the structure of the building and the building materials. Fortunately, experiments indicate that they need not be known with great accuracy to get reasonable results with the Bayesian grid update method. For example. reasonable position determination is possible using the parametric model developed in Section 3 under different circumstances.

Note again, that, in practice, a weighted sum of this distribution and a uniform distribution should be used to account for the appearance of random outliers in the measurements.

## 7. Some Extensions and Some Limitations

The observation model is based on data collected in many positions for initiators and responders. As a result, it is a kind of average and does not apply exactly to any one particular combination of initiator and responder positions. We could compensate exactly for biased distance measurements if we had a complete model of a building, including wall and floor thicknesses as well as their electrical properties at radio frequencies. This is obviously not realistic. A well-tuned observation model is the best we can do *without* such detailed modelling.

It is not only building materials that have remarkably high relative permittivities. The human body does as well. Some tissues have relative permittivities as high as 40 to 60, which, not surprisingly, comes close to that of water (ε=85 at 5 GHz and room temperature) [31]. The relative permittivity of lungs is lower, perhaps 20 to 40 depending on how much air they contain [31]. The distance added by an obstacle of thickness *d* and relative permittivity ε is $d(\sqrt{\varepsilon }-1)$.

A person holding a phone at chest level adds about 1 to 1.5 m to the reported distances for responders that are behind them. This can easily be seen as one turns to face different directions when using an application that continuously reports FTM RTT measurements. Naturally this limits the position accuracy that can be attained. The impact of this effect is reduced if more than the minimum required number of APs is used.

## 8. Conclusions

Bayesian cell update has proven to be a promising method for estimating indoor position using FTM RTT distance measurements [21]. It requires an observation model that gives the conditional probability of observing a reported distance given an actual distance.

(1) The parameters of an observation model, to be used with the Bayesian cell update, may be fitted using scattergrams of reported distances versus actual distance. We have shown that scattergrams of real data indicate that the observation model fitting problem can be reduced from two dimensions to one.

(2) We have further observed that, perhaps surprisingly, a “double exponential” observation model fits real data well—particularly if we include a flat top with parameterized position.

(3) Generating test data involves knowing not only the positions of the responders (APs) but also that of the initiator. The effort required for manual determination of initiator positions can limit the scale of test data collection. We have shown here that “boot strapping” using results of a Bayesian grid update method as a proxy for the actual position works well.

(4) Indoors, reported distance measurements are biased to be mostly larger than actual distances. This bias can be detected. Unfortunately, a simple heuristic for directly compensating for the bias does not appear to lead to much better results (see Appendix A). As a result, it appears that the best approach is to use the Bayesian cell update method without this heuristic compensation.

(5) Overall, indoor positioning using FTM RTT has significant advantages over competing methods, including higher accuracy and the ability to work with lower spatial density of devices (see Appendix C).

## Figures and Tables

**Figure 1 sensors-20-04027-f001:**
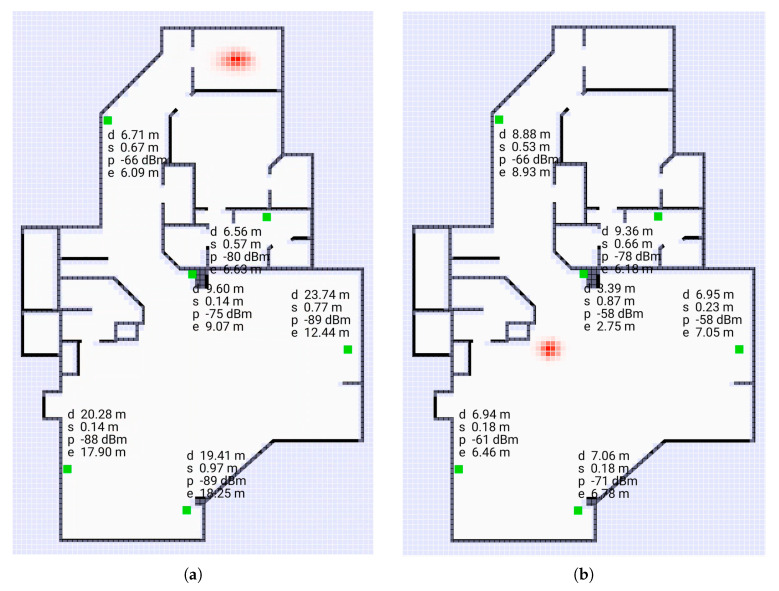
Screen shots of application using Bayesian grid update method as a “heat map” (from [29]). The cells here are 0.25 m on a side. The green spots are the responders. (**a**): initiator not far from centroid of responders, (**b**): initiator outside convex hull of responders. The “hot spot” (red) in (**b**) is larger than the one in (**a**). Note that the paths between initiator and responder typically passes through one or more walls or other obstacles. The grid update method uses the observation model developed here.

**Figure 2 sensors-20-04027-f002:**
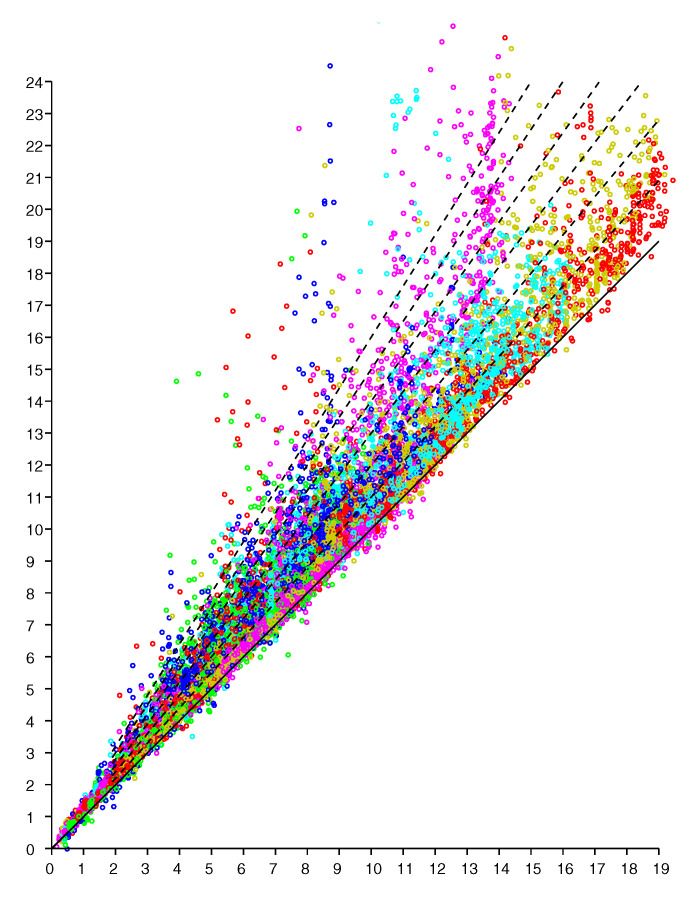
Scattergram of reported distance (vertical axis—meters) versus actual distance (horizontal axis—meters). Results from six responders color coded. (Dashed lines have slope 1.0, 1.1, 1.2 *…* 1.6 ).

**Figure 3 sensors-20-04027-f003:**
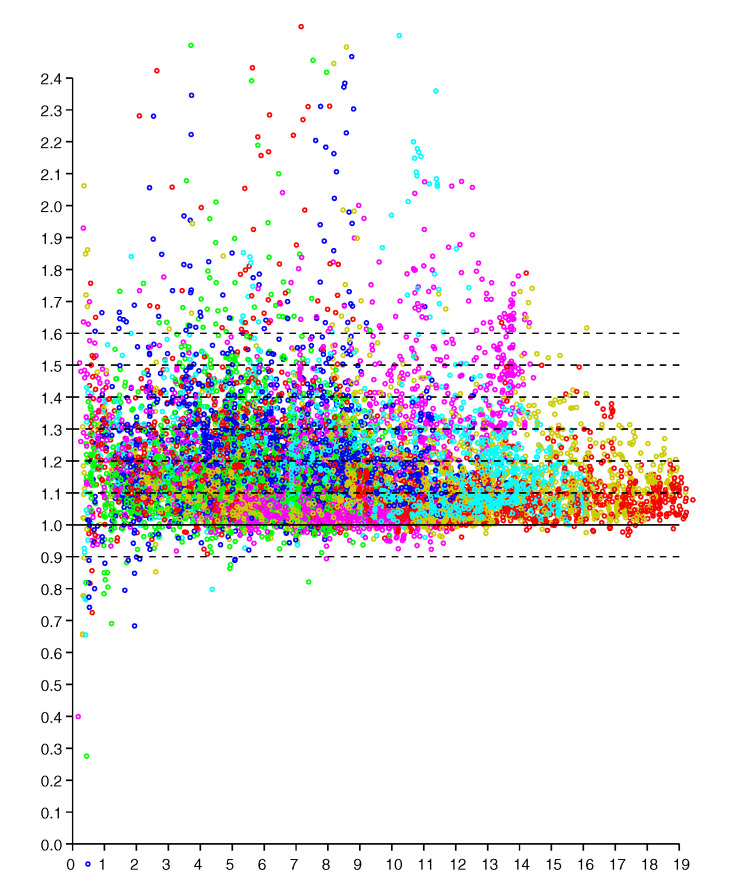
Scattergram of *ratio* of reported to actual distance (vertical axis) versus actual distance (horizontal axis—meters). Results from six responders color coded. (Dashed lines are shown for ratios of 0.9, 1.0 1.1, 1.2 *…* 1.6).

**Figure 4 sensors-20-04027-f004:**
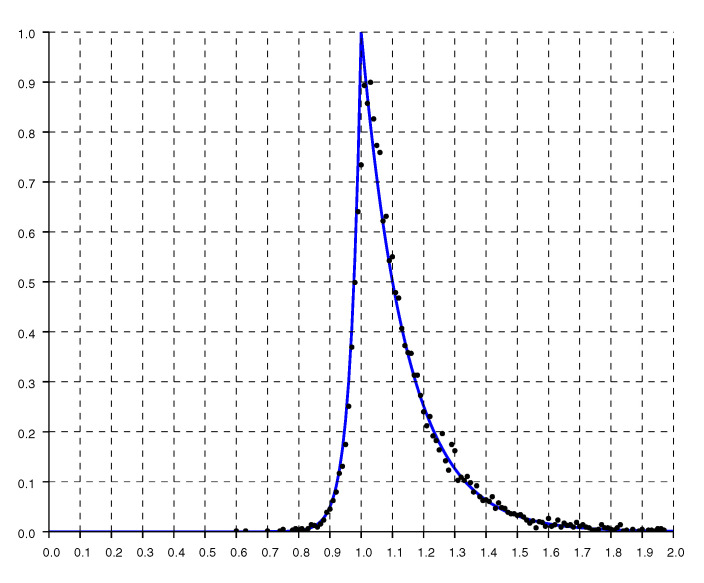
Histogram of the ratio of reported to actual distance based on about 10,000 measurements (scaled so that the peak equals one). The solid line is the “double exponential” parametric fit.

**Figure 5 sensors-20-04027-f005:**
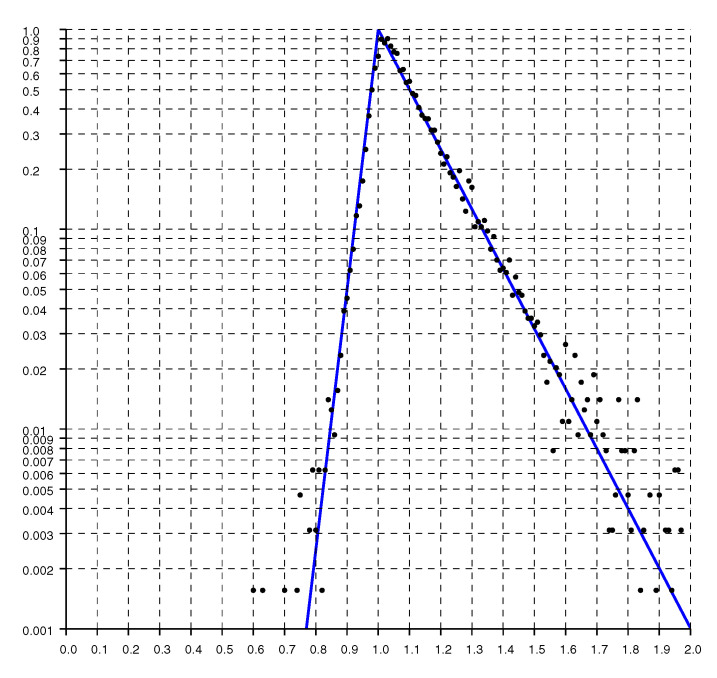
Histogram of the ratio of reported distances to actual distance on a logarithmic scale. The solid line is the “double exponential” parametric fit.

**Figure 6 sensors-20-04027-f006:**
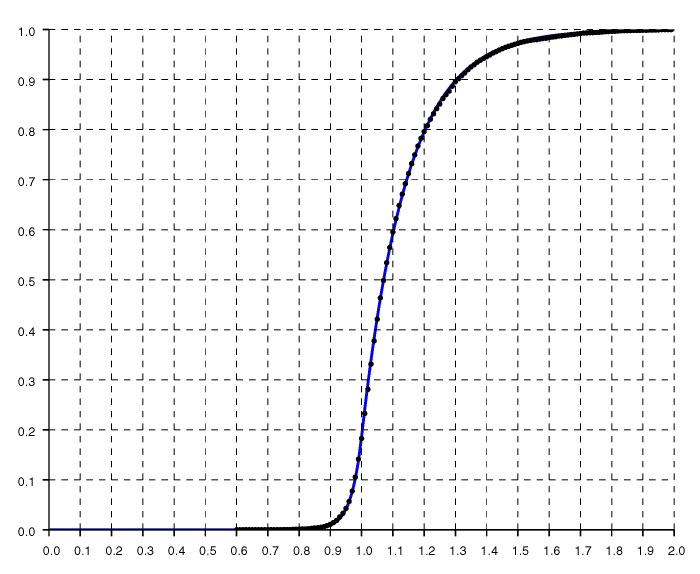
Normalized cumulative histogram of the ratio of reported to actual distance. The solid line is the “double exponential” parametric fit.

**Figure 7 sensors-20-04027-f007:**
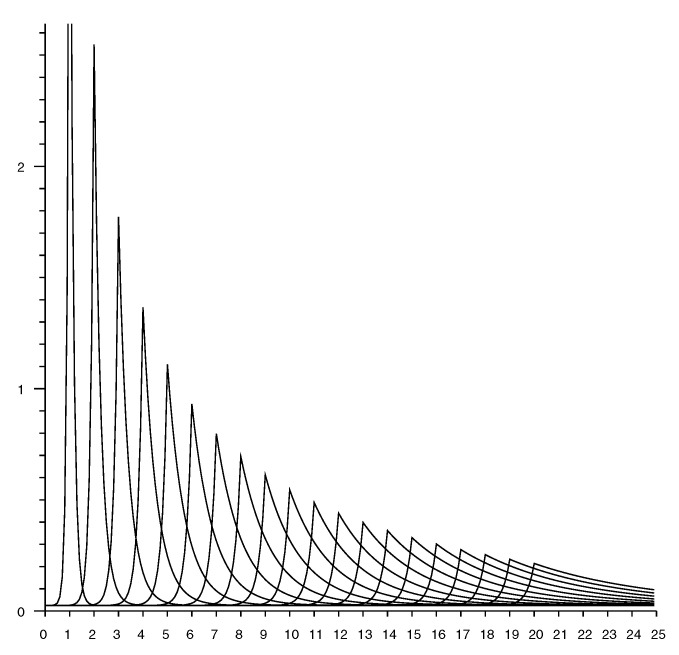
Conditional probability of observing a distance measurement (horizontal axis—meters). Each curve corresponds to a different value for the *actual* distance. Curves shown for actual distance of 1 m, 2 m, *…*, 20 m.

**Figure 8 sensors-20-04027-f008:**
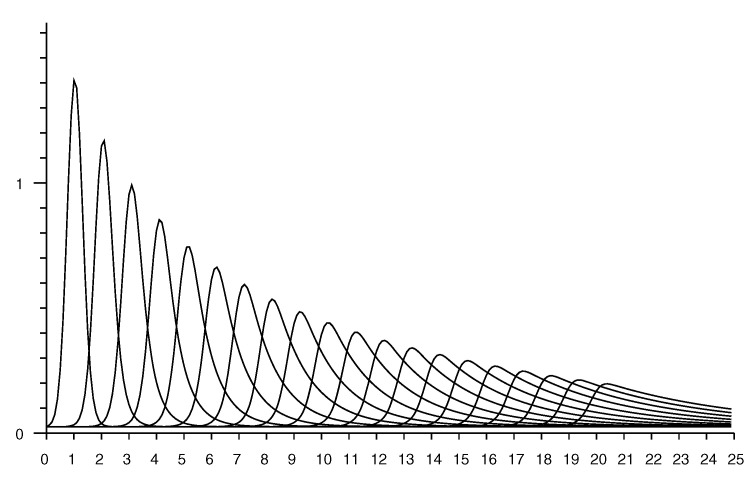
Conditional probability of observing a distance measurement (horizontal axis—meters) taking into account random measurement error with st. dev. of 0.25 m. Each curve corresponds to a different value for the *actual* distance. Curves shown for actual distance of 1 m, 2 m, *…*, 20 m.

**Figure 9 sensors-20-04027-f009:**
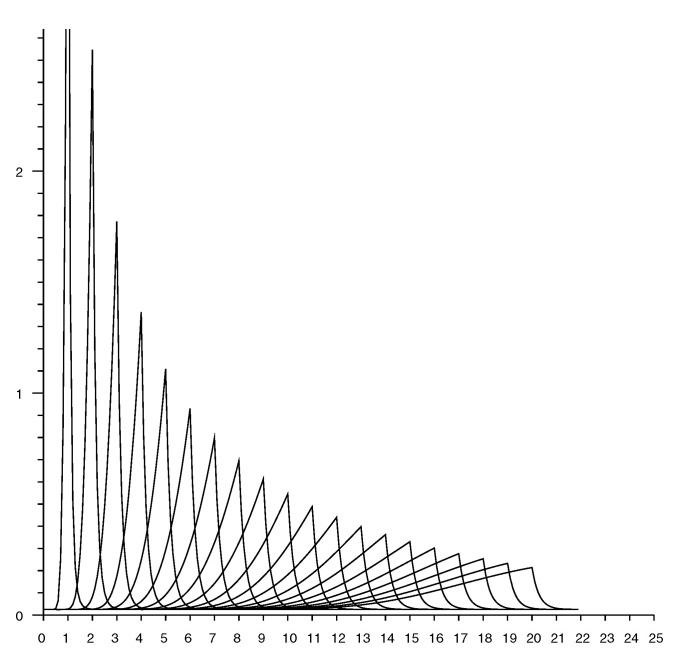
Rate vectors used in updating the Bayesian grid (horizontal axis actual distance—meters). Each curve corresponds to a different value for the *reported* distance. Curves shown for reported distance of 1 m, 2 m, *…*, 20 m.

**Figure 10 sensors-20-04027-f010:**
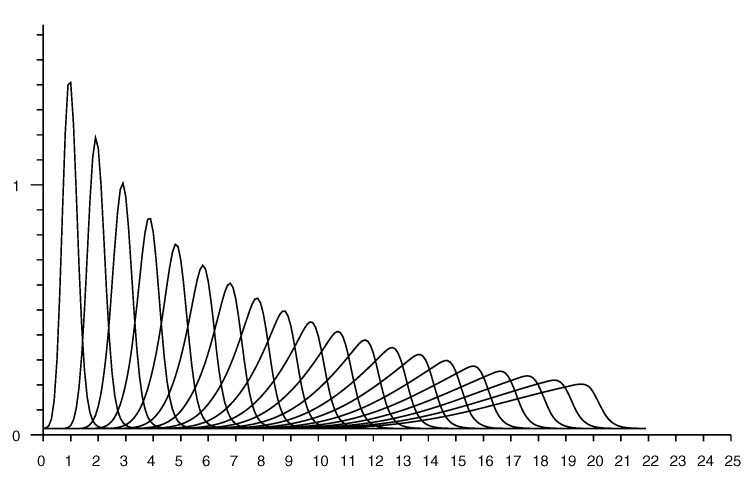
Rate vectors used in updating the Bayesian grid (horizontal axis actual distance—meters) taking into account measurement error with st. dev. of 0.25 m. Each curve corresponds to a different value for the *reported* distance. Curves shown for reported distance of 1 m, 2 m, *…*, 20 m.

**Figure 11 sensors-20-04027-f011:**
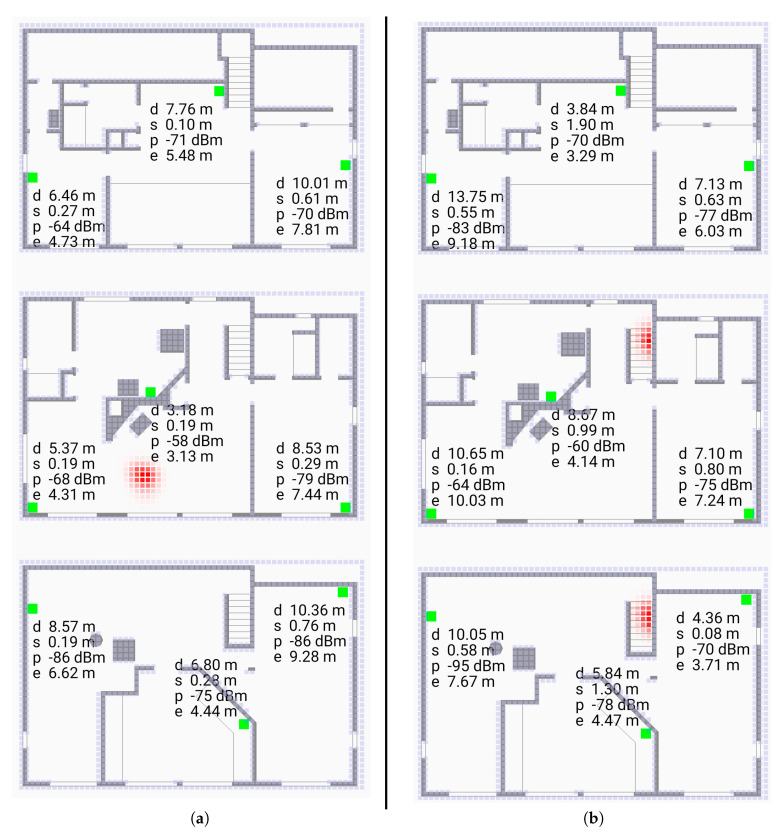
Screen shots of application using Bayesian grid update method as a “heat map” (from video [30]). The voxels are 0.2 m on a side. The green spots are the responders. (**a**) (left subfigure) initiator in middle of the middle floor (**b**) (right subfigure) initiator on stairs between floors. Note that the paths between initiator and responder in most cases pass through one or more walls or floors. The Bayesian grid update method uses the observation model developed here.

**Figure 12 sensors-20-04027-f012:**
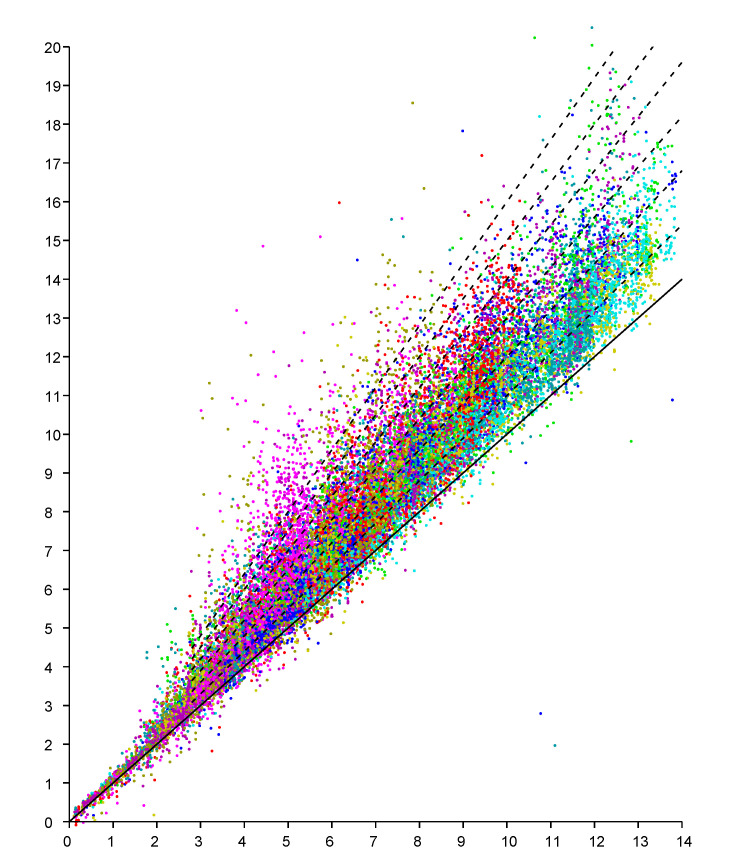
Scattergram of reported distance (vertical axis—meters) versus actual distance (horizontal axis—meters). Results from nine responders color coded. (Dashed lines have slope 1.0, 1.1, 1.2 *…* 1.6 ).

**Figure 13 sensors-20-04027-f013:**
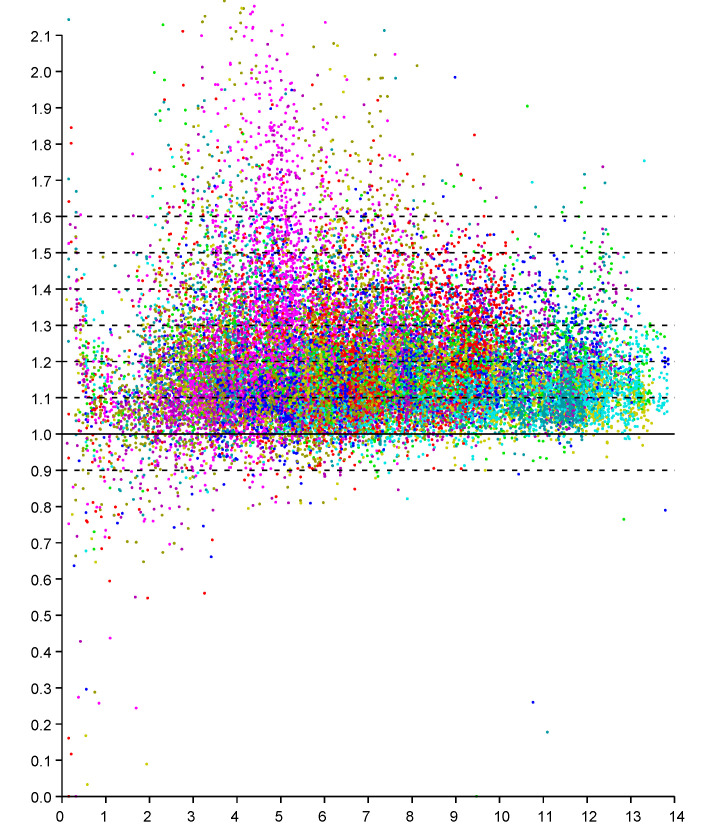
Scattergram of ratio of reported to actual distance (vertical axis) versus actual distance (horizontal axis—meters). Results from nine responders color coded. (Dashed lines are shown for ratios of 0.9, 1.0 1.1, 1.2 *…* 1.6).

**Figure 14 sensors-20-04027-f014:**
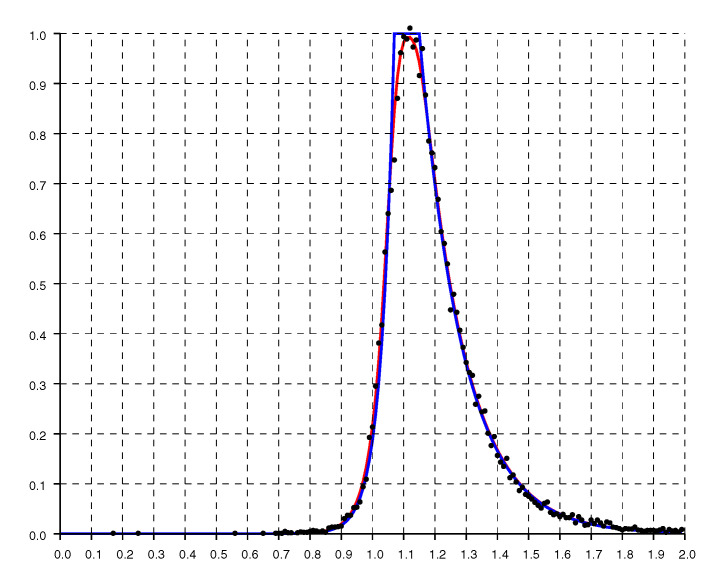
Histogram of the ratio of reported to actual distance based on about 20,000 measurements (scaled so that the peak equals one). The solid blue line is the “double exponential” with flat top parametric fit. The red curve is a smoothed version that takes into account measurement error with st. dev. of 0.25 m at 10 m.

**Figure 15 sensors-20-04027-f015:**
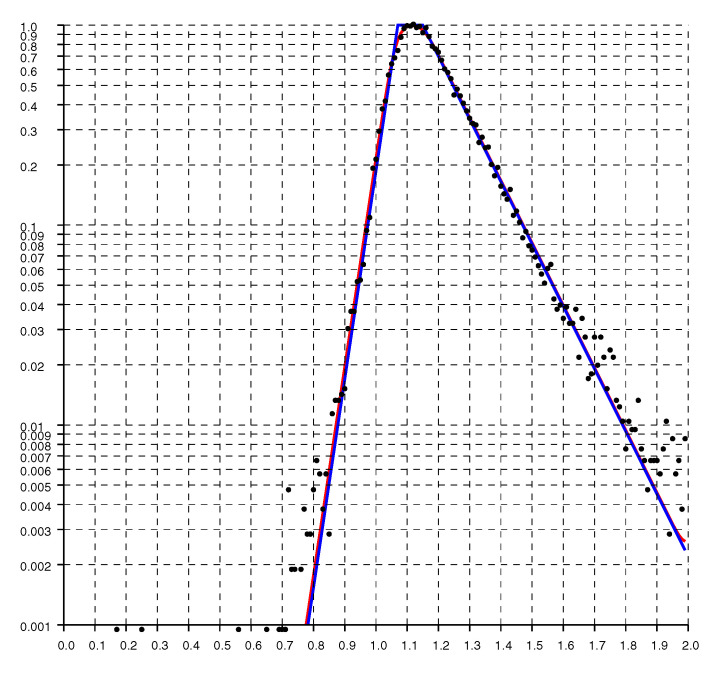
Histogram of the ratio of reported distances to actual distance on a logarithmic scale. The solid blue line is the “double exponential” with flat top parametric fit.

**Figure 16 sensors-20-04027-f016:**
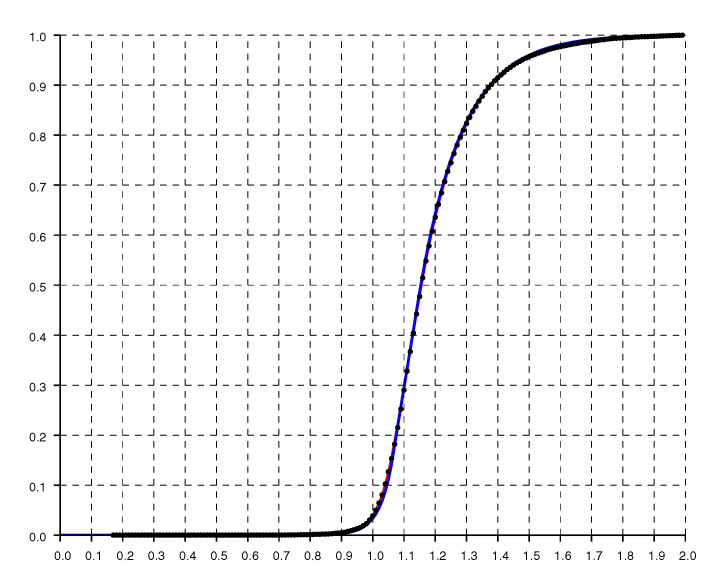
Normalized cumulative histogram of the ratio of reported to actual distance. The solid blue line is the “double exponential” with flat top parametric fit.

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
