# Peer review of "Observation Model for Indoor Positioning"

_sensors, 2020, doi:10.3390/s20144027_

Round 1

Reviewer 1 Report

This paper presents an observation model for indoor positioning based on a simple double exponential data fitting method. The paper is well-written and well-oganized, the method is properly developed and justified. A few minor comments are as follows:

  1. While it says the experiments were conducted in a single level of a house of many obstacles, it is unclear on what is the geometry of the initiator and responder. Did the signal cross walls?
  2. Related references should be provided, especially on signal propagation model-based methods (e.g., path loss model). 
  3. It would be more convincing to compare the positioning performance of the proposed model with existing models??

Author Response

This paper presents an observation model for indoor positioning based on a simple double exponential data fitting method. The paper is well-written and well-oganized, the method is properly developed and justified.

Thank you for your feedback and kind comments. I appreciate your effort in reviewing. Note: Changes in the text are marked in red.
Note: The numbering of references and figures below is that in the revised version.

A few minor comments are as follows:

While it says the experiments were conducted in a single level of a house of many obstacles, it is unclear on what is the geometry of the initiator and responder.

Sorry if this wasn't clear. Fig. 1 shows the floorplan of the house. The green spots are the positions of the responders. The initiator (cellphone) was carried around the house as shown in the video referenced in the caption of Fig. 1, namely:
https://people.csail.mit.edu/bkph/movies/FTM_RTT_wandering_2D.mp4
Fig. 1 shows two screenshots from that video (Measurements were taken in over 10,000 positions as the initiator was carried around the house). In response to the comment, I have added "The green spots are the positions of the responders." to the captions and made other minor changes to make this clearer.

Did the signal cross walls?

Yes, because of the layout of the house, many of the measurements were along lines that pass through obstacles - walls in particular. This accounts for the bias in the measured distances versus the actual distances. Most measured "distances" are significantly longer than the actual distances. To help explain this, I have added to the caption of Fig. 1:
"Note that the paths between initiator and responder typically pass through one or more walls or other obstacles." (and a similar addition to the caption of (the new) Fig. 17).

Prompted by this comment, I've added a second example that makes this more obvious. Section 6 (new) is for the "3-D" (multi-level residence) case what section 2 is for the "2-D" (one level) case. In the revised version of the paper, Fig. 11 shows the layout of a three-level house, where again the responders are the little green squares. These are screenshots from the video referenced in the caption of Fig. 11:
https://people.csail.mit.edu/bkph/movies/FTM_RTT_wandering_3D.mp4
In this case, actually *most* of the measurements go through walls and floors (often two or more).

Related references should be provided...

I apologize for the dearth of reference in this version. This was in part due to the notion that I should not duplicate much of the text and references from the earlier paper (reference [1]). But one can't reasonably assume that readers of this paper will go back to the earlier one. So I copied over some of the more relevant text and references from it. As a result, I have added about thirty (perhaps too much now?) references.

...especially on signal propagation model-based methods (e.g., path loss model).

Yes, thank you for the recommendation. I have added Appendix B which shows that signal strength itself is not really useful for determining the distance between initiator and responder, although this is not something new. Fig. 17 illustrates the "ill-posedness" of inversion from signal strength to distance.

I also added several references specifically related to material properties (relative permittivity and conductance) at GHz frequencies, and some on path loss (in particular references 25, 27, 28,29, 26, and 24).

It would be more convincing to compare the positioning performance of the proposed model with existing models??

There are "multi-lateration" methods that work in large open spaces
(now referenced in the bibliography - references 5, 6, 7, 8, 17 etc).
But, as pointed out in the paper:
"Since there is significant bias in the reported distances, one should not expect good results in "multi-lateration"
if one treats reported values as if they were the actual distances."
These methods do not work well in real residences and office buildings like the ones discussed here, because the measured distances are so different from the actual distances (bias). For example, several methods described on my web page http://people.csail.mit.edu/bkph/ftmrtt_location
including: reduction to linear equations, using pseudo-inverse, least-squares, gradient descent, brute force grid search, etc. did not work in this environment. The closest I came to something useful is shown in the video
https://people.csail.mit.edu/bkph/movies/FTM_RTT_naive_obs_model_3D.mp4
and that is not all that exciting. Hence not described in the paper.

I didn't reference most of this because (i) when things don't work at all, quantitative comparisons are not possible. (ii) the paper is already "too long", (iii) There are several examples of such failures and it would more than double the length of the paper if the justification for many arbitrary choices in implementation would have to be checked out and defended (iv) such a comparison could be the basis of future work (if funding for negative results could be found), although it seems more useful to put effort into methods that work :-). I am not saying that there may not be some way to get some modification of, for example, particle filters or Kalman filters to work well enough, but I wasn't able to see a way to do it (and have my doubts). Anyway, I have added in section I:

"There are several ``multi-lateration'' methods that work in relatively open environments." and
"The problem is harder in a typical residence where signals pass through walls and floors."

Also, there is a new appendix comparing the density of responders needed with the density of beacons needed in an alternate, beacon-based approach to indoor positioning.

Reviewer 2 Report

This paper presents and analyses an Observation Model for Indoor positioning for the IEEE 802.11mc WiFi standard. This work produces some interesting and useful results and provides helpful hints regarding the Observation model and its application.

However there are some drawbacks that must be looked after in order to improve the paper such that it can be useful for its readers.

  1. The main weakness of the paper is that it does not put the presented work in context with other related efforts in the field of Indoor positioning.

This can be done by extending the Background section and by including and commending on a number of state-of-the-art works in the field. As it is now there is only Ref [1] that the interested reader can refer to.

  1. On the same line with item No. 1 above, the References section must contain much more references as the field of Indoor positioning has attracted much interest during the last 5 years.
  2. In one case there is some suggestion as to what could be done for the method to improve estimation (page 11, line 6 from the top). The author mentions “to put all the blame on the largest measurement”. Probably he should try to investigate what happens if a few measurements of large value are omitted.
  3. page 10, line 4 from the top,: please clarify “vary with circumstances” in sentence “These biases increase with distance but vary with circumstances”
  4. page 10, line 13 from the top,: the sentence “…because the “forces” resulting from the biases will tend to be in different directions and partially cancel” needs clarification. For example, what happens if the point is very close to the border of the convex hull? In that case, distances from the corners that are far away from the initiator are large compared to the ones close to the initiator.
  5. For reasons of appearance of the work, Figures 7 to 10 must have a vertical scale as well.
  6. The TODO considerations at the end of the References must be erased.

Author Response

This paper presents and analyses an Observation Model for Indoor positioning for the IEEE 802.11mc WiFi standard. This work produces some interesting and useful results and provides helpful hints regarding the Observation model and its application.

Thank you for your feedback and kind comments. I appreciate your effort in reviewing. I hope I've been able to address all your comments adequately. Note: Changes in the text are marked in red. Note: The numbering of references and figures below is that in the revised version.

However there are some drawbacks that must be looked after in order to improve the paper such that it can be useful for its readers.

1. The main weakness of the paper is that it does not put the presented work in context with other related efforts in the field of Indoor positioning.

This can be done by extending the Background section and by including and commending on a number of state-of-the-art works in the field. As it is now there is only Ref [1] that the interested reader can refer to.

I apologize for the dearth of reference in this version. This was in part due to the notion that I should not duplicate much of the text and references from the earlier paper (reference [1]). But one can't reasonably assume that readers of this paper will go back to the earlier one. So I copied over some of the more relevant text and references from it. As a result, I have added about thirty references.

2. On the same line with item No. 1 above, the References section must contain much more references as the field of Indoor positioning has attracted much interest during the last 5 years.

Yes, agreed. Now, references 3--13 and 14--21 are about indoor positioning. Some text has been added, including in section 1:
"There has been considerable interest in developing the ability to accurately localize position indoors where GPS can not be used (references)"

3. In one case there is some suggestion as to what could be done for the method to improve estimation (page 11, line 6 from the top). The author mentions “to put all the blame on the largest measurement”.
Probably he should try to investigate what happens if a few measurements of large value are omitted.

Thank you for the suggestion. Indeed, with the bias - and its spread - increasing in proportion to distance, it is wise to place less emphasis on measurements with large distances. This happens automatically in the Bayesian update since the conditional probabilities (Fig. 7 and Fig. 8) get more spread out and drop off with larger distances.

But in response to the comment, omitting some of the contributions from far away responders is certainly an option and I have added some words to that effect to what is now Appendix A. One limitation in practice is that one will want to use as few APs as possible to keep costs down, so typically there won't be many more than the minimum number (3 for the 2-D case and 4 for 3-D case) within range. In the example illustrated in this paper, the density of responders is relatively high, since this was needed to make the proxy measurement of the initiator position accurate enough. For example, in the 2-D case (section 2, Fig. 1) six responders were used when three would have been enough for indoor positioning, and in the 3-D case (section 5, Fig. 11), nine responders were used, when four would have been enough. Thanks to this suggestion I have added:

"A related idea (suggested by one of the reviewers) is to actually {\it omit} one or more of the longest distance measurements since they are less reliable. This can help if the density of responders is high enough that they are not all needed to solve for the position (more than 3 responders within range in 2-D, 4 responders within range in 3-D). Note that in the examples shown here, the density of responders happens to be higher than normally needed for indoor positioning because more responders were used for the collection of data for the fitting of probability densities, as mentioned above. In practice, there often wouldn't be enough responders within range to make it possible to discard some measurements. Note, by the way, that the Bayesian update method discounts contributions from far away responders since the conditional probabilities for those are more spread out and smaller (see Fig.~\ref{SLICES:OBSERVATION:2D} and
Fig.~\ref{SLICES:OBSERVATION:SMOOTHED:2D})."

4. page 10, line 4 from the top,: please clarify “vary with circumstances” in sentence “These biases increase with distance but vary with circumstances”

The bias depends on the arrangements of materials that the line connecting the initiator to the responder passes through. So the error is not a fixed one at a given distance. So the bias will be relatively small in a relatively open arrangement in a lecture hall, but large in a typical home. I have added (note that this has been moved to Appendix A):

"The magnitude of the effect depends on what obstacles the line connecting the initiator to the responder passes through."

5. page 10, line 13 from the top,: the sentence “…because the “forces” resulting from the biases will tend to be in different directions and partially cancel” needs clarification. For example, what happens if the point is very close to the border of the convex hull? In that case, distances from the corners that are far away from the initiator are large compared to the ones close to the initiator.

Yes, in this simple analogy, the effects of the "forces" depend not only on the direction but also the magnitude. Generally speaking closer responders have more of an effect than distant once. But it's not all that simple. Consider, for example, the bottom left corner of the right subfigure of Fig. 14 of reference [1]. It shows that even very near to one responder the error ellipses can get very large when the directions to *other* responders are similar rather than spread out. Further, looking at Fig. 12 in reference [1] makes it clear that being close to a responder is not an unmitigated blessing: the areas that satisfy the constraints get spread out (larger) and have odd shapes (even bimodal). So somewhat surprisingly, being close to a responder is not always that helpful. So the statement in the paper is of a heuristic nature: it's generally true, but it does not address the details or "edge cases". I don't have any further great insights on this approximate reasoning so rather than speculate I just added the following:

"(see also error ellipses in Fig.~14 in \cite{Horn2020})"

6. For reasons of appearance of the work, Figures 7 to 10 must have a vertical scale as well.

OK. (These are probability distributions and so integrate to one).

7. The TODO considerations at the end of the References must be erased.

Sorry about that. (Not sure how they got included since I thought they were past the \end{document})